# Evaluation of Five International HBV Treatment Guidelines: Recommendation for Resource-Limited Developing Countries Based on the National Study in Nepal

**Sundar Khadka** [1,2,*,†], **Roshan Pandit** [1,†], **Subhash Dhital** [1], **Jagat Bahadur Baniya** [1], **Surendra Tiwari** [1], **Bimal Shrestha** [1], **Sanjeet Pandit** [1], **Fumitaka Sato** [2], **Mitsugu Fujita** [2], **Mukunda Sharma** [1], **Ikuo Tsunoda** [2,*] **and Shravan Kumar Mishra** [1]

1   National Public Health Laboratory (NPHL), Department of Health Services, Ministry of Health and Population, Teku, Kathmandu 44-600, Nepal; roshanpandit2050@gmail.com (R.P.); dhital.subhash@gmail.com (S.D.); baniyajagat571@gmail.com (J.B.B.); tiwarisuren16@gmail.com (S.T.); mech.bimal@gmail.com (B.S.); sanjitpandit@gmail.com (S.P.); mukubattarai@yahoo.com.sg (M.S.); shravan.nepal@gmail.com (S.K.M.)

2   Department of Microbiology, Kindai University Faculty of Medicine, Osakasayama, Osaka 589-8511, Japan; fsato@med.kindai.ac.jp (F.S.); mfujita47@gmail.com (M.F.)

*   Correspondence: cls.sundar@gmail.com (S.K.); itsunoda@med.kindai.ac.jp (I.T.)

†   These authors contributed equally to this work.

**Abstract:** Hepatitis B virus (HBV) infects the liver, causing cirrhosis and cancer. In developed countries, five international guidelines have been used to make a decision for the management of patients with chronic HBV infection. In this review, since the guidelines were established by clinical and epidemiological data of developed countries, we aimed to evaluate whether (1) HBV patient profiles of developing countries are similar to developed countries, and (2) which guideline can be applicable to resource-limited developing countries. First, as an example of the most recent data of HBV infections among developing countries, we evaluated the national HBV viral load study in Nepal, which were compared with the data from other developing countries. In Nepal, the highest number of patients had viral loads of 20–2000 IU/mL (36.7%) and belonged to the age group of 21–30 years; HBV epidemiology in Nepal, based on the viral loads, gender, and age groups was similar to those of not only other developing countries but also developed countries. Next, we reviewed five international HBV treatment guidelines of the World Health Organization (WHO), American Association for the Study of Liver Diseases (AASLD), National Institute for Health and Care Excellence (NICE), European Association for the Study of the Liver (EASL), and Asian Pacific Association for the Study of the Liver (APASL). All guidelines require the viral load and alanine aminotransferase (ALT) levels for decision making. Although four guidelines recommend elastography to assess liver cirrhosis, the WHO guideline alternatively recommends using the aspartate aminotransferase (AST)-to-platelet ratio index (APRI), which is inexpensive and conducted routinely in most hospitals. Therefore, in resource-limited developing countries like Nepal, we recommend the WHO guideline for HBV treatment based on the viral load, ALT, and APRI information.

**Keywords:** chronic human hepatitis; DNA viruses; elastography; *Hepadnaviridiae*; viral diseases

## 1. Introduction

### 1.1. Hepatitis B Virus (HBV) Infection

Hepatitis B virus (HBV) is an enveloped DNA virus that belongs to the family *Hepadnaviridae*, genus *Orthohepadnavirus*. HBV infects the liver and can cause chronic hepatitis with necrosis and inflammation, which sometimes results in liver cirrhosis and cancer [1]. Hepatitis B is a serious infectious disease and one of the major causes of morbidity and mortality in tropical and subtropical countries. In the world, at least two billion people are infected with HBV, and approximately 260 million people are chronic carriers [2,3]. The prevalence of HBV infection based on the hepatitis B surface antigen (HBsAg) categorizes countries into four groups: the high (≥8%), high-intermediate (5–7%), low-intermediate (2–4%), and low (<2%) HBV infection prevalence countries.

HBV is transmitted by percutaneous or mucosal exposure to infected body fluids, including blood. Following HBV transmission, the viral infection progresses from the acute stage to the chronic stage [4]. The acute stage of HBV infection is characterized by acute liver inflammation and hepatocellular necrosis. The chronic stage is defined as persistent infection with detectable HBsAg for longer than 6 months in the presence or absence of evidence showing active viral replication, hepatocellular injury, or inflammation [5].

### 1.2. HBV Diagnosis

HBV infection is diagnosed by the detection of either the viral antigens, including the HBsAg and hepatitis B envelope antigen (HBeAg), or anti-HBV antibodies, including anti-HBsAg (anti-HBs), anti-hepatitis B core antigen (anti-HBc), and anti-HBeAg (anti-HBe) antibodies, in blood samples [5]. The presence of the HBsAg or HBeAg in blood indicates HBV infection and/or active HBV replication. The presence of anti-HBe antibody indicates spontaneous improvement with a decline in the viral replication. The presence of anti-HBs and anti-HBc antibodies indicates past infection [6,7].

The HBV viral loads (HBV DNA concentration) quantified by real-time polymerase chain reaction (PCR) have also been used to evaluate disease progression and to distinguish an active HBeAg-negative disease from an inactive chronic infection, which helps in decision-making for subsequent treatment or monitoring [7]. The viral load test is useful in the decision-making of treatment along with other parameters, such as alanine aminotransferase (ALT), aspartate aminotransferase (AST)-to-platelet ratio index (APRI), and liver fibrosis [7–9]. APRI is used as a non-invasive method to assess cirrhosis in resource-limited settings; liver fibrosis can be detected by biopsy or magnetic resonance elastography [5]. There are five major international guidelines for treating patients with HBV infection: The World Health Organization (WHO), American Association for the Study of Liver Diseases (AASLD), National Institute for Health and Care Excellence (NICE), European Association for the Study of the Liver (EASL), and Asian Pacific Association for the Study of the Liver (APASL). All the guidelines include viral loads. In this review, we aimed to propose a guideline for chronic HBV treatment for developing countries, since the five international guidelines, which were established by the clinical setting and data of developed countries, may not be applicable to resource-limited developing counties. First, we will use the data of the first national HBV viral load study in Nepal, as the most recent data representative of developing counties. Then, we will discuss epidemiology of chronic HBV infection based on the viral loads, gender, and age groups, by comparing Nepal with other countries. Lastly, we will review the five international HBV guidelines to evaluate which guideline can be applicable for resource-limited countries like Nepal.

## 2. HBV Epidemiology in Nepal, Compared with Other Developing Countries

Historically, the five international guidelines have been established based on clinical and epidemiological data obtained from developed countries, for example, AASLD from the USA and NICE from the UK. There have been few studies regarding HBV profile characterization in developing countries. In this section, we will characterize clinical and epidemiological profiles of chronic HBV

infection in developing countries by comparing the most recent national study data in Nepal with those in other developing countries.

## 2.1. The First National HBV Viral Load Study in Nepal

In Nepal, the investigation for HBV infection started in 1983 with a single blood marker HBsAg [10]. Although Nepal belongs to the low prevalence countries [11], nearly 260,000 individuals (0.9%) have been chronically infected with HBV without knowing their own HBV infection [10]. Here, we evaluated the data of the first national HBV viral load study, which was conducted in Nepal as a retrospective study at the HIV Reference Unit of the National Public Health Laboratory (NPHL), Kathmandu, Nepal. In the study, 300 samples were analyzed from patients who were diagnosed with HBV infection from January to June 2016 [12]. A limitation of the study was that the samples were only from Kathmandu, which may not represent the local areas in Nepal, although the sample size was adequate based on the formula recommended in prevalence studies [13,14].

## 2.2. Viral Load

In Nepal, the patients with the viral loads of 20–2000 IU/mL were the most common (36.7%) (Figure 1A). This was similar to the report by Iregbu et al., who found that Nigeria had the highest number of patients with viral loads of 20–2000 IU/mL [15]. The percentages of patients with viral loads of 2000–20,000 IU/mL and more than 20,000 IU/mL in Nepal were 11.7% and 19.3%, respectively. HBV DNA was not detected in 29.3% of patients despite their positive HBsAg. This may be due to the higher sensitivities of the HbsAg rapid test than the viral load test [16]. The level of HBV DNA also depends on the stages of viral infection and decreases faster than that of HBsAg due to its shorter half-life [16].

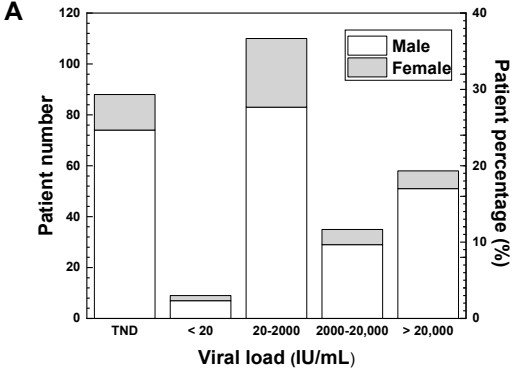 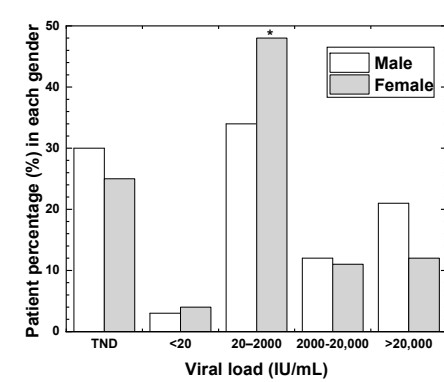

**Figure 1.** (**A**) Levels of hepatitis B virus (HBV) DNA in plasma among the hepatitis B surface antigen (HbsAg)-positive patients. Each column represents the number (left y axis) and percentage (right y axis) of HBV patients (open, male; shaded, female) at each range of viral loads. The percentage of the HBV patients was calculated as follows: % = (Male + Female patient number at each viral load range) X100/ 300 (=total patient number of HBV samples). The highest number of patients was in the viral load range of 20–2000 IU/mL. Among 300 patients tested for the viral loads, 244 patients (81.3%) were male and 56 patients (18.7%) were female. (**B**) Percentage of HBV patients at each viral load range in each gender (open, male; shaded, female). The percentage of patients with viral loads of 20–2000 IU/mL was highest in both males and females, and significantly higher in females (48.2%) than in males (34.0%) (* $p < 0.05$, $\chi^2$ test). The percentage of the HBV patients in each gender was calculated as follows: %  = (Patient number at each viral load range in males or females) X100/ [244 samples (males) or 56 samples (females)]. HBV DNA was quantified by real-time polymerase chain reaction (PCR) using the COBAS® AmpliPrep/COBAS® TaqMan® System (Roche Diagnostic, Pleasanton, NJ, USA). The linear range of the standard curve was 20–$1.7 \times 10^8$ IU/mL; the HBV DNA levels less than 20 IU/mL were described as <20 IU/mL. When HBV DNA was undetectable, it was described as the target not detected (TND).

## 2.3. Gender

When the percentages between males and females among 300 samples were compared, 244 samples (81.3%) were male and 56 samples (18.7%) were female (male:female ratio, 4.4:1). This is similar to other studies from Nepal by Gupta et al., Shrestha et al. and Bhatta et al. reporting high male-to-female ratios of HBV infection (2.5:1, 3.2:1, 9.0:1, respectively) [17–19]. The high male-to-female ratios of HBV infection have also been reported in recent epidemiological studies in Ghana (2.2:1), China (1.5:1), and Pakistan (2.1:1) as well as retrospective data of 1982–1998 in the USA (1.8:1) [20–23]. In Nepal, males travel abroad and engage in unsafe sexual activities more than females, which could explain the male preponderance of HBV infection. Another reason may be due to cultural aspects of males who are thought to be involved in outdoor activities, while females are restricted to do only household activities in Nepal [18]. This may also be due to the sociological factors of Nepal; more economic resources are available to males who can afford medical examination than females.

In Nepal, the percentage of patients with viral loads of 20–2000 IU/mL was significantly higher in females (48.2%) than in males (34.0%) ($p < 0.05$, Figure 1B), although there were no significant gender differences in the other ranges of viral loads. Regarding viral loads ranging from 20 to 2000 IU/mL, Iregbu et al. have reported no significant difference in the percentage of patients between genders, i.e., 45.9% (102/222) in females and 47.3% (210/444) in males in Nigeria [15].

## 2.4. Age and Vaccination

In Nepal, the highest number of patients belonging to the age group of 21–30 years was 98 (32.7%) (Figure 2A); this study was similar to the studies conducted in Pakistan (the highest number in the age group of 21–30 years) and South Africa (the highest numbers in the age groups of 20–29 and 30–39 years) [6,24]. On the other hand, in two studies conducted in Nigeria, the highest population of HBV patients belonged to the age group of 31–40 years [15,25]. The highest incidence of the age group of 21–30 years in Nepal could be due to the spread of HBV among adolescents by horizontal transmission, as suggested by Shrestha et al. [18]. Although the risk of HBV infection mediated by drug injection is less than 10% in general, a horizontal transmission has been observed as a major cause of chronic HBV infection in the countries with low prevalence of HBV infection (<2%), including Nepal [26].

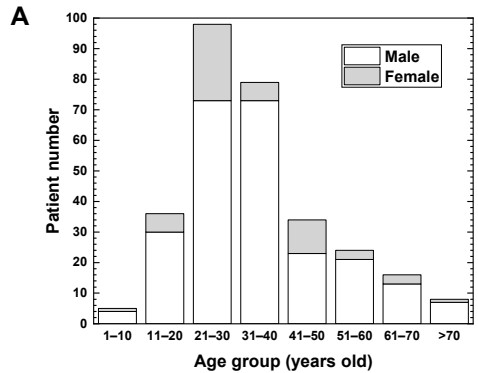 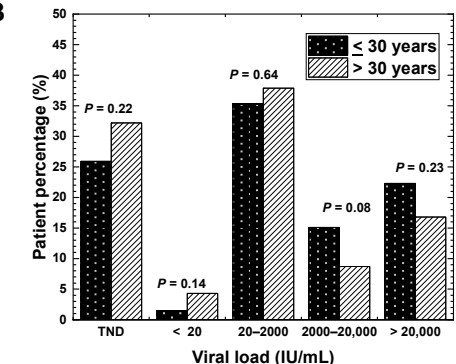

**Figure 2.** (**A**) HBV patient distribution at different age ranges. Each column represents the number of HBV patients (open, male; shaded, female) at each age range. The highest number of patients was the age range 21–30 years old, followed by 31–40 years old. (**B**) HBV patient percentages in different viral load ranges at two age-groups. The patients were categorized into two age-groups; 139 patients were 30 years old or below (filled bar), and 161 patients were above 30 years old (hatched bar). The percentages in two age-groups had no significant difference in any viral load ranges. $p$ values were calculated by the $\chi^2$ test. The HBV patient percentages in each age-group were calculated as follows: % = (Number of patients <30 years-old or patients >30 years-old at each viral load range) X100/[139 samples (<30 years-old) or 161 samples (>30 years-old)].

In Nepal (Figure 2A), the least number of patients belonging to the age group of 1–10 years were five (1.7%). The low incidence of HBV-positive children and young adults (<20 years old) in Nepal could be explained by the fact that approximately 20 years have passed since the HBV vaccination started in 2002 in Nepal, which was a relatively earlier introduction of HBV vaccination than other resource-limited countries, including South Africa in 2007, and Pakistan in 2009 [24,27,28]. The introduction of HBV vaccination has been shown to alter the age-group distribution of the HBV patients. Goldstein et al. have demonstrated that, in the USA, the incidence of hepatitis B decreased after the introduction of the HBV vaccination in 1988 [23].

*2.5. Age and Viral Load*

Patients older than 30 years of age have a poor prognosis of chronic HBV infection [5]. For example, in South Africa, the viral loads were higher in the age group of above 30 years than in the age group of 30 years or below [15]. In Nepal (Figure 2B), the patients were categorized into two age groups: 139 patients (46.3%) were 30 years old or below and 161 patients (53.7%) were above 30 years old. Unexpectedly, however, the proportions of patients belonging to the two age groups were not significantly different in any viral load ranges.

## 3. Five International HBV Treatment Guidelines

In the above section, we have evaluated the clinical and epidemiological profiles of chronic HBV infection in Nepal, comparing viral load, gender, age and vaccination with other developing countries. Overall, the profiles in Nepal were similar to those in not only other developing countries but also developed countries [22–24,27]. Thus, clinically and epidemiologically, chronic HBV data profiles on viral load are similar between developed and developing countries. Although more comprehensive comparative studies among countries are ideal, using the data on other factors/parameters, such as HBeAg, anti-HBe, ALT, and APRI, no data sets including all the parameters were available in Nepal or other developing countries. Here, purely scientifically (if one does not consider the economical settings of developing countries whose resources are limited, compared with developed countries), the international guidelines for HBV treatment established in developed countries can also be applicable for developing countries. In this section, we have summarized five international guidelines for the HBsAg-positive patients. This is the first review article that compares the five guidelines. By comparing the guidelines, we found that (1) in all the guidelines, the treatment strategies are similarly determined by the levels of HBV DNA, ALT, liver fibrosis, and age, and (2) only the WHO guideline recommends the APRI to evaluate liver fibrosis as an alternative to elastography. [9,15,29–31].

*3.1. World Health Organization (WHO, 2015)*

The WHO recommends treatment for all patients with cirrhosis as well as the patients with HBV DNA levels of more than 20,000 IU/mL and persistently abnormal ALT levels without cirrhosis, particularly those older than 30 years of age [5] (Table 1). Continuous monitoring is required for the patients under 30 years old with HBV DNA levels of more than 20,000 IU/mL and persistently normal ALT levels as well as for those with HBV DNA levels of 2000–20,000 IU/mL. In contrast, treatment is not recommended for those with HBV DNA levels of less than 2000 IU/mL and persistently normal ALT levels without clinical evidence of cirrhosis [5]. The WHO guideline recommends the APRI to assess the presence of cirrhosis (APRI > 2) in resource-limited settings. In addition to viral loads, ALT levels are required to determine the treatment of HBV infection.

**Table 1.** WHO treatment indications for chronic HBV-infected patients.

| Viral Load (IU/mL) | ALT Level | Age (Years) | Recommendation |
|---|---|---|---|
| >20,000 | Abnormal | >30 (in particular) | Treatment |
| | Normal | ≤30 | Monitoring |
| 2000–20,000 | Abnormal | >30 | Monitoring |
| <2000 | Normal | All | No treatment |

Abbreviations: ALT, alanine aminotransferase; HBV, hepatitis B virus; WHO, World Health Organization.

### 3.2. American Association for the Study of Liver Diseases (AASLD, 2018)

The AASLD recommends treatment for all patients with cirrhosis as well as HBeAg-positive patients with the HBV DNA levels of more than 20,000 IU/mL and persistently increased ALT levels (>two times the upper limit of normal) without cirrhosis; or persistently increased ALT levels without cirrhosis, particularly those older than 40 years of age [32] (Table 2). This guideline also recommended treating patients with HBV DNA levels between 2000 and 20,000 IU/mL with persistently increased ALT levels, particularly those older than 40 years of age. Continuous monitoring is required for patients with the HBV DNA levels of more than 20,000 IU/mL, and with normal ALT levels if HBeAg is positive. In contrast, treatment is not recommended for those with HBV DNA levels of less than 2000 IU/mL and normal ALT levels without cirrhosis [32]. The AASLD guideline recommends liver biopsy, elastotography and liver fibrosis biomarkers (FIB-4 or FibroTest) to assess liver fibrosis.

**Table 2.** AASLD treatment indications for chronic HBV-infected patients.

| Viral Load (IU/mL) | ALT Level | HBeAg | Recommendation |
|---|---|---|---|
| >20,000 | >2 × ULN | Positive | Treatment |
| | Increased | Positive | Treatment (if age > 40 years) |
| | Normal | Positive | Monitoring |
| 2000–20,000 | Increased | Both | Treatment (if age > 40 years) |
| <2000 | Normal | Negative | No treatment |

Abbreviations: ALT, alanine aminotransferase; AASLD, American Association for the Study of Liver Diseases; HBV, hepatitis B virus; HBeAg, hepatitis B envelope antigen; ULN, upper limit of normal.

### 3.3. National Institute for Health and Care Excellence (NICE, 2013)

The NICE, UK, recommends treatment for patients with HBV DNA levels of more than 20,000 IU/mL and abnormal ALT levels in two consecutive tests at 3-month intervals; treatment can be initiated regardless of the patient's age or the extent of liver disease [33] (Table 3). The treatment should be initiated for the patients aged 30 years or older with HBV DNA levels of more than 2000 IU/mL and abnormal ALT levels (≥30 IU/L in males and ≥19 IU/L in females) in two consecutive tests at 3-month intervals. In the patients younger than 30 years of age with HBV DNA levels of more than 2000 IU/mL and abnormal ALT levels in two consecutive tests at 3-month intervals, treatment is initiated only if there is evidence of severe hepatitis or fibrosis. The NICE guideline requires transient elastography and serum ALT levels in addition to viral loads to assess chronic hepatitis B and fibrosis.

**Table 3.** NICE treatment indications for chronic HBV-infected patients.

| Viral Load (IU/mL) | ALT Level | Age (Years) | Recommendation |
|---|---|---|---|
| >20,000 | Abnormal | All | Treatment |
| 2000–20,000 | Abnormal | >30 | Treatment |
| | Abnormal | ≤30 | Treatment (if severe hepatitis or fibrosis) |
| <2000 | Normal | All | No treatment |

Abbreviations: ALT, alanine aminotransferase; HBV, hepatitis B virus; NICE, National Institute for Health and Care Excellence.

### 3.4. European Association for the Study of the Liver (EASL, 2017)

The EASL recommends treatment for patients with HBV DNA levels of more than 20,000 IU/mL and abnormal ALT levels [34] (Table 4). It also suggests treatment for the patients if their HBV DNA levels exceed 2000 IU/mL, ALT levels are elevated, and active necrosis/inflammation in the liver is observed. Patients under 30 years of age with high HBV DNA levels and no evidence of liver disease are not immediately treated but should be kept on follow-up [34]. In the EASL guidelines, the indications for treatment are based on the combinations of three criteria: the HBV DNA levels, ALT levels, and severity of liver diseases as determined by biopsy or elastography.

**Table 4.** EASL treatment indications for chronic HBV-infected patients.

| Viral Load (IU/mL) | ALT Level | Liver Biopsy | Recommendation |
|---|---|---|---|
| >20,000 | >2 × ULN | Not required | Treatment |
| | Normal | Not required | Monitoring (if age < 30 years) |
| 2000–20,000 | Increased | Active necrosis/liver inflammation | Treatment |
| <2000 | Normal | Not required | No treatment |

Abbreviations: ALT, alanine aminotransferase; HBV, hepatitis B virus; EASL, European Association for the Study of the Liver; ULN, upper limit of normal.

### 3.5. Asian Pacific Association for the Study of the Liver (APASL, 2015)

The APASL recommends treatment for patients based on their HBV DNA levels, ALT levels and severity of liver disease, as well as age, health status, family history, and hepatic manifestations [3] (Table 5). Patients with chronic HBV infection without cirrhosis can also be treated if they have persistently increased ALT levels (>two times the upper limit of normal) for at least 1 month and HBV DNA levels of more than 20,000 IU/mL (with positive HBeAg) or HBV DNA levels of more than 2000 IU/mL (with negative HBeAg) [3]. According to this guideline, patients with compensated cirrhosis and the HBV DNA levels of less than 2000 IU/mL should also be considered for treatment, even if their levels of ALT are normal. Liver cirrhosis is assessed by a non-invasive method by using Fibroscan or APRI, although liver biopsy is a recommended method.

**Table 5.** APASL treatment indications for chronic HBV-infected patients.

| Viral Load (IU/mL) | ALT Level | Liver Disease | Recommendation |
|---|---|---|---|
| >20,000 | >2 × ULN | Pre-cirrhotic chronic hepatitis | Treatment (if HBeAg-positive) |
| 2000–20,000 | Increased | Cirrhosis/inflammation | Treatment (if age > 35 years) |
| >2000 | >2 × ULN | Pre-cirrhotic chronic hepatitis | Treatment (if HBeAg negative) |
| <2000 | Normal | Compensated cirrhosis | Treatment |
| | Normal | – | Monitoring |

Abbreviations: ALT, alanine aminotransferase; HBV, hepatitis B virus; APASL, Asian Pacific Association for the Study of the Liver; ULN, upper limit of normal.

*3.6. Treatment of HBV Infection Based on Five International Guidelines*

Five international guidelines also described the treatment of HBV infection. The main goal of antiviral therapy is to decrease the morbidity and mortality related to chronic HBV infection by delaying the progression of cirrhosis, reducing the incidence of hepatocellular carcinoma, and improving long-term survival. All guidelines recommend either interferon (IFN)-based therapy including peg-IFN, or nucleos(t)ide analogs (NAs)-based therapy including lamivudine, telbivudine, entecavir, adefovir, tenofovir, and emtricitabine. There are six therapeutic agents (peg-IFN-α-2a, lamivudine, telbivudine, entecavir, adefovir, and tenofovir) for adults and five therapeutic agents (IFN-α-2b, lamivudine, entecavir, adefovir, and tenofovir) for children [32]. All guidelines recommended tenofovir-based therapy, which is also recommended in Nepal [35]. In Nepal, antiviral medications started with lamivudine since 2001 [10] and currently include peg-IFN and other NAs, such as entecavir and tenofovir [36]; peg-IFN has been used in limited cases due to its side effects. During the treatment, the WHO recommends at least annual monitoring of ALT, HBsAg, HBeAg, HBV DNA levels, and APRI scores for liver cirrhosis assessment.

## 4. Guideline Proposal for Resource-Limited Developing Countries

In resource-limited developing countries like Nepal, most institutes/clinicians are unable to conduct complete diagnostic tests for HBV carriers, such as viral loads, ALT, APRI, elastography, HBsAg, and HBV genotypes. Thus, until recently physicians had to decide therapeutic strategies based on their clinical experience with limited diagnostics tests. Under such circumstances, there are no standard national guidelines for the management of chronic HBV infection in most resource-limited countries, including Nepal. A review of five international guidelines for the management of chronic HBV infection indicates that all guidelines require the evaluation of liver fibrosis to determine treatment. In the absence of liver fibrosis, treatment decisions are generally made based on the HBV DNA levels, ALT levels, and age. Although most guidelines recommend the use of elastography to evaluate the presence of liver cirrhosis, the WHO guideline alternatively recommends the use of the APRI, which are inexpensive compared with other medical imaging technologies and less invasive than liver biopsy. Therefore, in resource-limited countries including Nepal, we recommend following the WHO guidelines with information on the APRI, ALT levels and HBV DNA levels that can be used to determine whether to initiate treatment or to continue monitoring (Figure 3).

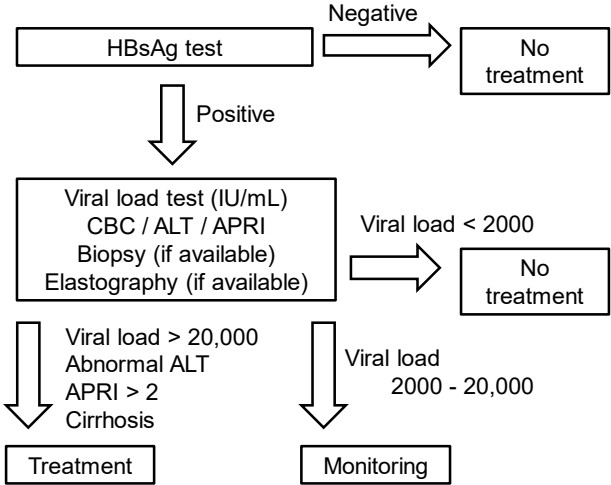

**Figure 3.** Proposed algorithm for treatment of chronic hepatitis B virus (HBV) infection for resource-limited countries based on the viral loads. First, clinicians conduct laboratory tests of the HbsAg in blood, diagnose HBV infection, and test the viral loads, alanine aminotransferase (ALT), complete blood counts (CBC), and the aspartate aminotransferase-to-platelet ratio index (APRI) as well as elastography, if it is available. Since the World Health Organization (WHO) guideline recommends the use of the APRI to assess the presence of cirrhosis (APRI > 2) in resource-limited settings, we propose the use of the APRI, ALT levels, and viral load information to determine treatment of HBV infection.

## 5. Conclusions

Based on the data of the first national HBV viral load study in Nepal, we have discussed the epidemiology of HBV infection. Then, we reviewed five international guidelines for treating patients with HBV infection. Among the guidelines, we recommend the WHO guideline for resource-limited developing countries, since the WHO guideline requires the data of APRI, ALT, and viral loads, all of which can be tested in these countries. The establishment of the guideline in line with the situation in each country will help physicians to determine whether to initiate treatment or to continue the monitoring of HBV patients.

**Author Contributions:** S.K., R.P., and I.T. designed this review manuscript and went through a literature review. S.K., F.S., M.F., and I.T. prepared this manuscript. S.K., J.B.B., S.T., B.S., S.D. and S.P. collected the data. S.K., R.P., M.S., S.K.M. and I.T. conducted data analyses. All authors have read and agreed to the published version of the manuscript.

**Funding:** This work was supported by grants from the National Institute of General Medical Sciences COBRE Grant (8P20 GM 103433, I. Tsunoda), and the Japan Society for the Promotion of Science [(Grant-in-Aid for Scientific Research (C) KAKENHI Grant Number JP20K07455)] (I. Tsunoda), and Research Program on Emerging and Re-emerging Infectious Diseases from the Japan Agency for Medical Research and Development (AMED) under grant number 19fk0108084h0801 (I. Tsunoda), and Novartis Pharma Research Grants (I. Tsunoda).

**Acknowledgments:** We express our acknowledgments to all the staff of the NPHL, Teku, Kathmandu, as well as members of the Department of Microbiology, Kindai University Faculty of Medicine, Ah-Mee Park, Seiich Omura, Namie Sakiyama, Aoshi Katsuki, Yumina Nakamura, and Felicia Lindeberg. This work was supported by the Ministry of Education, Culture, Sports, Science and Technology, Japan, through the Monbukagakusho (MEXT) Scholarship (SK).

**Conflicts of Interest:** The authors declare no conflict of interest.

## Abbreviations

| | |
|---|---|
| AASLD | American Association for the Study of Liver Diseases |
| ALT | Alanine aminotransferase |
| APASL | Asian Pacific Association for the Study of the Liver |
| APRI | Aspartate aminotransferase-to-platelet ratio index |

| EASL | European Association for the Study of the Liver |
| HBcAg | Hepatitis B core antigen |
| HBeAg | Hepatitis B envelope antigen |
| HBsAg | Hepatitis B surface antigen |
| HBV | Hepatitis B virus |
| NICE | National Institute for Health and Care Excellence |
| PCR | Polymerase chain reaction |
| TND | Target not detected |
| WHO | World Health Organization |

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
