# Peer review of "Evaluation of Five International HBV Treatment Guidelines: Recommendation for Resource-Limited Developing Countries Based on the National Study in Nepal"

_pathophysiology, doi:10.3390/pathophysiology27010002_

Round 1
Reviewer 1 Report
This review is a significant contribution to possible treatment of patients with chronic HBV-infection in developing countries especially in Nepal. First, the authors present data on the prevalence of HBV-infection in various age groups of Napalese people and report the different levels of HBV DNA. They present the different guidelines established for developed countries, some requiring expensive diagnostic tools e.g. elastography. The authors have to be congratulated that they present data supporting WHO recommendations which allows developing countries to use resource saving indeces for decision of therapy in their countries.
Author Response
We appreciate the reviewer's comment.
Reviewer 2 Report
The paper is well written and addresses significant problems in the management of HBV chronic infection in developing countries.
In addition to the algorithm about the decision of the treatment or not, it could be useful that the authors indicate the drugs available in their country and how the patients are monitored during the treatment.
Author Response
We appreciate the comment and now included a separate paragraph on treatment and monitoring of HBV with two new references as follows:
New
lines: 260-273, page 7
3.6 Treatment of HBV infection based on five international guidelines
Five international guidelines also described the treatment of HBV infection. The main goal of antiviral therapy is to decrease the morbidity and mortality related to chronic HBV infection by delaying the progression of cirrhosis, reducing the incidence of hepatocellular carcinoma, and improving long-term survival. All guidelines recommended either interferon (IFN)-based therapy including peg-IFN, or nucleos(t)ide analogs (NAs)-based therapy including lamivudine, telbivudine, entecavir, adefovir, tenofovir, and emtricitabine. There are six therapeutic agents (peg-IFN-2a, lamivudine, telbivudine, entecavir, adefovir, and tenofovir) for adults and five therapeutic agents (peg-IFN-α-2b, lamivudine, entecavir, adefovir, and tenofovir) for children [32]. All guidelines recommended tenofovir-based therapy, which is also recommended in Nepal [35]. In Nepal, antiviral medications started with lamivudine since 2001 [10] and currently have included peg-IFN and other NAs, such as entecavir and tenofovir [36]; peg-IFN has been used in limited cases due to its side effects. During the treatment, the WHO recommends at least annual monitoring of ALT, HBsAg, HBeAg, HBV DNA levels, and the APRI for liver cirrhosis assessment.
new references
Ref 35: National centre for AIDS and STD control, Nepal. National HIV Testing and Treatment Guidelines 2017; 2017.
Ref 36: Masaki N, Shrestha PK, Nishimura S, Ito K, Sugiyama M, Mizokami M. Use of nucleoside analogs in patients with chronic hepatitis B in Nepal: A prospective cohort study in a single hospital. Hepatol Res. 2015;45(12):1163-9.
Reviewer 3 Report
Khadka et al. evaluated the clinical and epidemiological profiles of chronic HBV infection in Nepal analysing viral load, age and gender in 300 patients. Moreover, they evaluated 5 international HBV treatment guidelines with the aim to indentify the most suitable for their Country.
Major concerns
1) Despite the interesting topic, I can’t consider as an epidemiological study the analysis of only 300 samples from a Country with 260,000 infected subjects, especially if the criterion on the basis of which these serums were chosen is not defined.
2) HBeAg, anti HBe, ALT and most important the fibrosis stratification based on APRI (how many cirrhotic?) are missing
3) The detailed analysis of the 5 guidelines to be applied seems superfluous when it is known that the only possibility to evaluate liver fibrosis by non invasive test is the APRI index only
Author Response
Major concerns
1) Despite the interesting topic, I can’t consider as an epidemiological study the analysis of only 300 samples from a Country with 260,000 infected subjects, especially if the criterion on the basis of which these serums were chosen is not defined.
We appreciate the comment. As to the sample size, the following formula has been used for calculating the adequate sample size in prevalence studies (Naing et al., 2006; Pourhoseingholl et al., 2013).
n = Z2 P(1-P)
d2
where n = sample size, Z = Z statistic for a level of confidence, P = expected prevalence, and, d = precision
Based on the formula, the sample size required for the Nepalese study is n=240 with 99% confidence intervals. Thus, the sample size of the study is adequate. However, since the samples were collected from 10 institutes in Kathmandu, the study data may not represent HBV epidemiology of the local areas in Nepal. We have included this limitation with two new references as follows:
Page 3, lines 95-97
A limitation of the study was that the samples were only from Kathmandu, which may not represent the local areas in Nepal, although the sample size was adequate based on the formula recommended in prevalence studies [13, 14].
Ref 13: Naing L, Winn T, Rusli BN. Practical issues in calculating the sample size for prevalence studies. Arch Orofacial Sci. 2006;1:9–14.
Ref 14: Pourhoseingholi MA, Vahedi M, Rahimzadeh M. Sample size calculation in medical studies. Gastroenterol Hepatol Bed Bench. 2013;6(1):14-17.
2) HBeAg, anti HBe, ALT and most important the fibrosis stratification based on APRI (how many cirrhotic?) are missing
We agree with the comment; this is another limitation of this study, particularly when we assess whether clinical and epidemiological profiles of HBV infections are similar among countries. We have included the limitation as follows:
Old
Page 5, lines 178-179
Thus, clinically and epidemiologically, chronic HBV data profiles are similar between developed and developing countries.
New
Page 5, lines 180-184
Thus, clinically and epidemiologically, chronic HBV data profiles on viral load are similar between developed and developing countries. Although more comprehensive comparative studies among countries are ideal using the data on other factors/parameters, such as HBeAg, anti-HBe, ALT, and APRI, no data sets including all the parameters were available in Nepal or other developing countries.
3) The detailed analysis of the 5 guidelines to be applied seems superfluous when it is known that the only possibility to evaluate liver fibrosis by non invasive test is the APRI index only
We appreciate the comment. Our current manuscript is the first review article that compares the five major international guidelines for HBV management. By comparing the guidelines, we found that 1) the treatment strategies in all guidelines are similarly determined by the levels of HBV DNA, ALT, liver fibrosis, and age, and 2) only the WHO guideline recommends the APRT to evaluate liver fibrosis as an alternative to elastography. We clarified the point as follows:
Old
Page 5, lines 183-185
Basically, in all the guidelines, the treatment strategies are determined by the levels of HBV DNA, ALT, liver fibrosis, and age.
New
Page 5, lines 188-192
This is the first review article that compares the five guidelines. By comparing the guidelines, we found that 1) in all the guidelines, the treatment strategies are similarly determined by the levels of HBV DNA, ALT, liver fibrosis, and age, 2) only the WHO guideline recommends the APRI to evaluate liver fibrosis as an alternative to elastography.
Round 2
Reviewer 3 Report
Authors answered all my questions